# Effect of Alloying Elements on the Sharpness Retention of Knife Blades Made of High Carbon Martensitic Stainless Steels

**Dong Wu** [1,2,3] **, Qinyi Zhang** [1,2,3],* **and Wei Liu** [3]

1    School of Materials Science and Engineering, Wuhan University of Technology, Wuhan 430070, China; dongwu@whut.edu.cn
2    Yangjiang Branch, Guangdong Laboratory for Materials Science and Technology (Yangjiang Advanced Alloys Laboratory), Yangjiang 529500, China
3    Yangjiang Tuobituo Industrial Technology Research Institute Co., Ltd., Yangjiang 529500, China; lw@tuobituo.com
*    Correspondence: zhqy@whut.edu.cn; Tel.: +86-181-6226-2798

**Abstract:** Blades usually become blunt as the blade tip suffers wear during cutting, and improving the sharpness retention of steel blades has become an attractive prospect in various industries. In this study, blades were fabricated from commercial high carbon martensitic stainless steels (154CM, 440C and N690) with different contents of alloying elements. 154CM with higher Mo content demonstrated superior capability in sharpness retention to 440C and N690, although these steels exhibit similar chemical composition, carbide phases, microstructure and HRC hardness. Further investigations via SEM and nanoindentation indicated that the faster deterioration of sharpness in 440C and N690 may result from the fatigue peeling of the carbides, which was aggravated by the modulus mismatch between carbide particles and the martensitic steel matrix.

**Keywords:** sharpness; abrasive wear; fatigue wear; martensitic stainless steels





## 1. Introduction

Sharpness is one of the most important features that impacts the working performance of cutting tools (e.g., kitchen knives). It is well accepted that under most circumstances less force is required for a sharper blade edge in the cutting process. A knife sharpness evaluation (KSE) method has been developed based on this idea [1], and a critical force of 40N is suggested as a criterion to judge whether a knife needs to be sharpened. Usually, sharp blades are desired in various industrial applications [2] e.g., meat cutting [3], paper shearing [4], as well as personal care such as razor blades. According to previous studies [1,5,6], blade geometry is the most crucial factor that influences the cutting performance of blade (especially blade sharpness). To be more specific, the edge angle and the radius of the blade tip as well as the blade shape that extends from the blade tip down to the knife body could be mentioned as blade geometry. Marsot et al. [1] assessed knife sharpness using a cutting force measuring system, and found that the cutting force is proportional to the tangent of the blade half edge angle. This effect of blade edge angle on cutting force has also been verified by C.T. McCarthy et al. [6] via FEM calculation. Moreover, they also revealed that the cutting force increases with the increasing of blade tip radius. Besides, J.D. Verhoeven et al. [7] compared two types of edge geometry of blades as well as their cutting performance, and found a larger cutting depth for the thinner blade body under the same cutting load in a cutting cycle.

Derived from the conception sharpness, the retention of sharpness is introduced and discussed in the manufacturing industry [3,8]. The sharpness of a blade usually decays since the blade suffers abrasive processes during cutting, and the blade geometry turns from sharp to blunt as the wear accumulates. In fact, wear occurs when blades cut materials from hard paper, hair [9], and soft meat [10], and the more a blade is used, the blunter

it becomes. To prolong the service lifetime of blades, it is attractive to study the factors that affect the wear process of the blades and develop techniques accordingly to improve sharpness retention of blades. The mechanical properties of blade materials (e.g., hardness, wear resistance, fracture toughness), as demonstrated in previous studies [11–15], affect the wear process of blades. The capability of maintaining the sharpness of blade is greatly enhanced when the blade materials are strengthened. That is the reason why carbon was introduced into blade steels, in which hard carbon-soluted martensite phase were obtained after a special quenching treatment (i.e., fully austenitized at high temperature and then fast quenching to the room temperature). Recently, carbon was added up to high content (>1 wt%) in the martensitic stainless steels (MSSs), and carbide precipitates were formed in the steel matrix [16], thereby further strengthened this kind of steel. It has been determined that the sharpness of knives made of MSS 8Cr13MoV is sensitive to the size, shape, and quantity of carbides [17].

To prevent the corrosion often seen in kitchen knives made of carbon steels, stainless steel was developed by heavily alloying Cr, which forms a passivation layer [18] on the knife surface. Currently, MSSs have been intensively used in the kitchen knife fabrication due to their good performance in both sharpness retention and corrosion resistance [19]. More alloy elements (e.g., Mo and W) have been added in these MSSs to resolve localized pitting issues [20,21] that occur occasionally in the presence of chloride. The alloying elements Mo and Ni are also reported to improve the tensile properties of MSS 13Cr [22]. However, the effect of alloying elements (e.g., Mo and Cr) on the sharpness retention of MSS blade is not yet well investigated. It can facilitate the design and development of knives with better cutting performance in industries if the roles of alloying elements are well understood.

In this manuscript, three commercial high-carbon MSSs were properly heat treated, and fabricated into blades with the same geometry. The cutting performance, including sharpness and sharpness retention of these blades, were evaluated and compared using an international standard testing technique. The chemical composition, microstructure, and mechanical properties of these steels were investigated to reveal a possible cause of unexpected sharpness performance. It was found that proper alloying can increase the mechanical stability of carbides in martensitic matrix and improve the retention of blade sharpness.

## 2. Sample Processing and Experimental Testing

Commercial MSS plates were purchased as raw materials for knife fabrication: 154CM and 440C from Crucible Industry and N690 from Bohler-Uddenholm Steel Industry. The metallic elements of three high carbon MSSs (154CM, 440C, N690) were examined using X-ray fluorescence spectrometer (Zetium, Malvern PANalytical, Malvern, UK) and the carbon contents were measured using infrared absorption method (HIR-944B, Ningbo Yinzhou Jinrui Instrument Equipment Co. Ltd. Ningbo, China).

Before mechanical fabrication, these steel plates with the thickness of 5 mm were heat-treated in the procedure (Figure 1) as follows: at first homogenized at 840 °C for 60 min, then austenized at 1060 °C for 30 min, the steel plates were then quenched in oil for martensitic transformation, and tempered at 200 °C for 210 min to relieve internal stress.

X-ray Diffraction patterns of the heat-treated steel plates were collected in the 2θ range of 30–120° at a step rate of 2°/min using a X'Pert PRO MPD X-ray diffractometer (Malvern PANalytical, Malvern, UK) equipped with Co radiation (λ = 0.179021 nm). For hardness tests, the steel plates after heat treatment were cut into small pieces with the size of 10 by 10 mm, and one side of the samples were ground and polished into mirror surface. For the observation of microstructure, additional etch treatment was required. A typical etch operation is to soak the polished side of these steel samples in the etchant (5 g $FeCl_3$ + 10 mL 37% HCl + 85 mL $H_2O$) for 10–15 s. The microstructure and elemental composition of the phases in steel samples were characterized using backscattered imaging mode on the electron probe microanalyzer (EPMA, JXA-8230, JOEL Ltd., Tokyo, Japan). The amount

of each phase in the microstructure was estimated from its area occupied in the scanning electronic micrograph using image processing software Image Pro Plus.

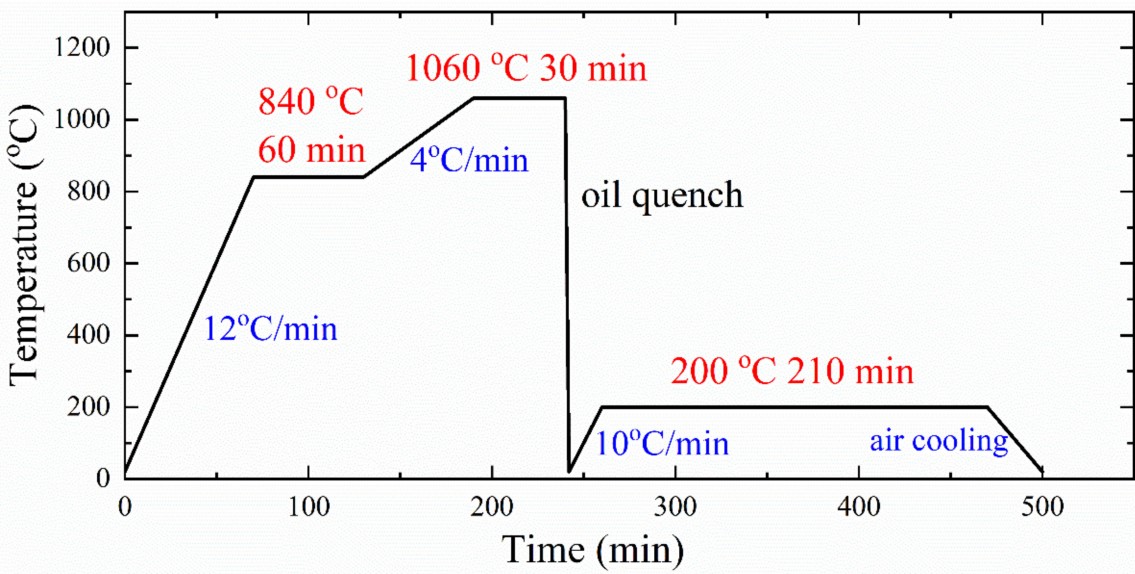

**Figure 1.** Heat treatment processes for the blade steels.

For the evaluation of blade sharpness, all of the steel plates were punched into knife-shape samples, as displayed in the inset of Figure 2. One long edge of the knife-shape sample was ground into sharp blade. The angle of blade tip is about 24 degree, and the round tip geometry was observed in Figure 2 after the sharpness test.

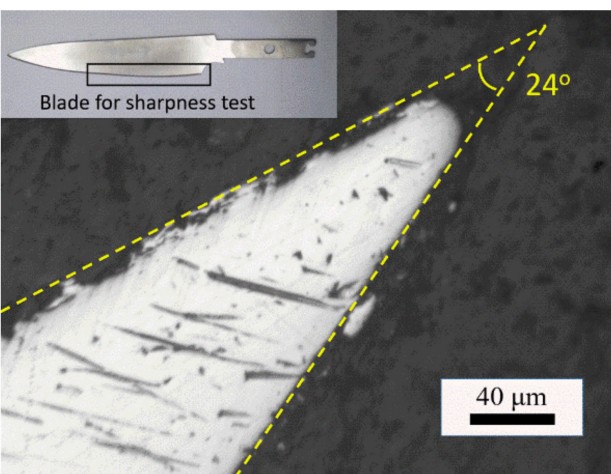

**Figure 2.** The blade edge for sharpness test on a sample of steel knife (inset) and the round tip geometry with the blade angle 24° after the sharpness test.

The cutting performance of stainless-steel blades were evaluated using a testing machine developed by CATRA [2,7,23] (see schematic diagram in Figure 3), and these tests followed the procedure of ISO8442-5:2004(E). Though multiple sharpness testing techniques [6,24–26] has been developed so far, and different properties such as cutting force [1,25], blade tip radius, and offset [17,26], KSE [3] or BSI [6,24] have been introduced as a measure of blade sharpness, the CATRA method is still preferred in this type of study [17]. The main reason for this is that both sharpness and its retention capability can be evaluated using a standardized testing technique and the blade edge gets worn by the abrasive particles contained in the paper cards. In a typical CATRA cutting test, a sharp steel blade cut stacks of paper cards in reciprocating movement (Stroke length

per cut = 40 mm, cutting speed = 50 mm/s) for 60 cycles under a normal load of 50N. The initial cutting performance (ICP) of a knife blade is defined to be the cumulative cutting depth of stacked cards in the first three cycles. The cumulative cutting depth of stacked cards in the 60 cycles is defined as total card cutting (TCC) for a blade, which evaluates the persistence of blade cutting performance. About 5 wt% $SiO_2$ particles are contained in card fibers (as shown in Figure 4a,b). The majority of these $SiO_2$ particles are smaller than 10 μm, and the friction between these particles and steel blade causes worn blade edge in the cutting test. Examples of blade edge before and after cutting test can be found in a knife made of 154CM stainless steel, as displayed in Figure 5a,b, respectively. The sharp edge before cutting test identified by red dotted line degraded into a round end in the green blanket zone, and abrasive grooves indicated by the red arrow on the round end were probably scratched by the $SiO_2$ particles in the paper cards.

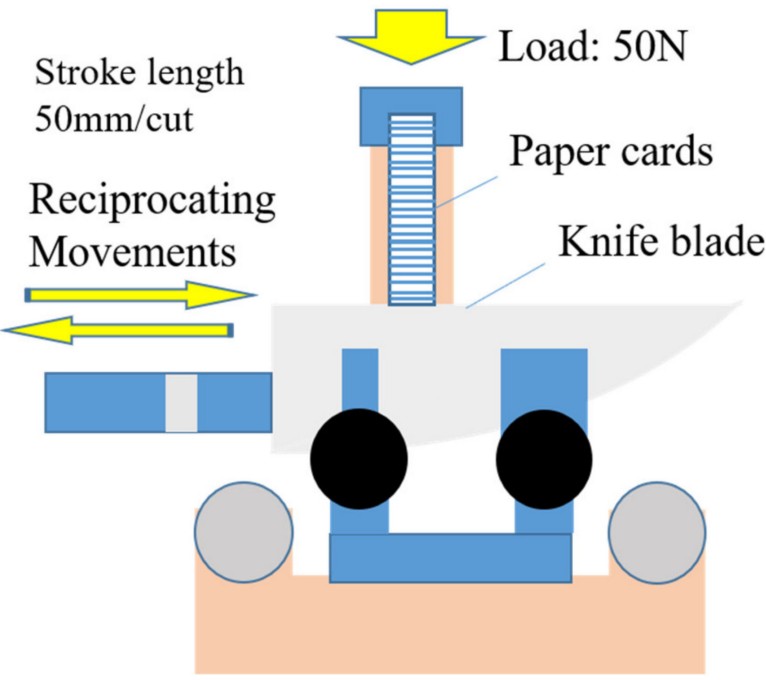

**Figure 3.** Schematic diagram of testing machine for the blade sharpness evaluation.

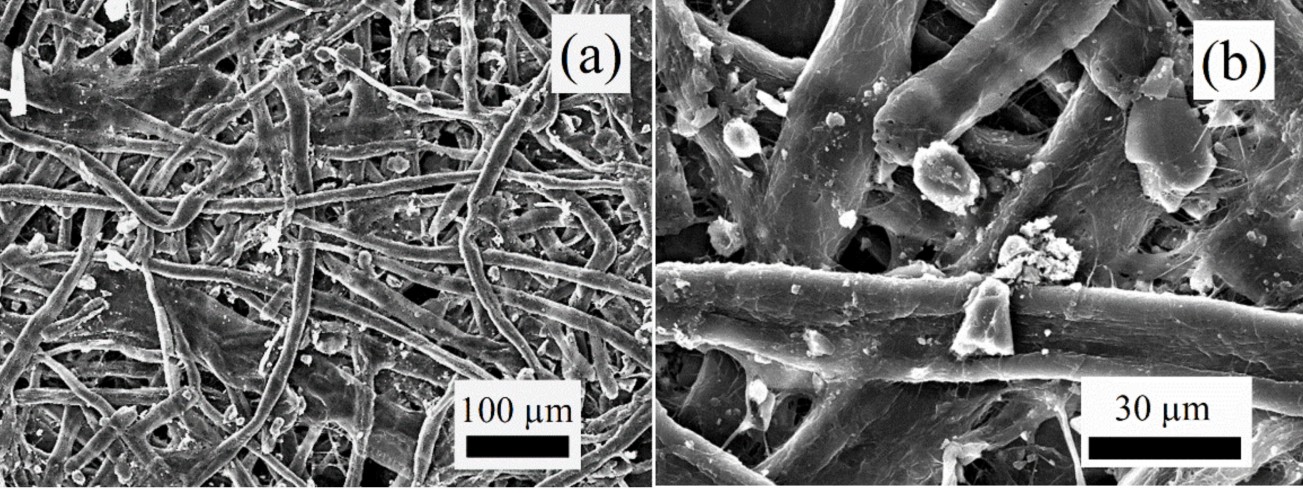

**Figure 4.** SEM micrographs of the paper cards used for blade sharpness test shows paper fibers with abrasive $SiO_2$ particles (**a**) and enlarged view of those $SiO_2$ particles (**b**).

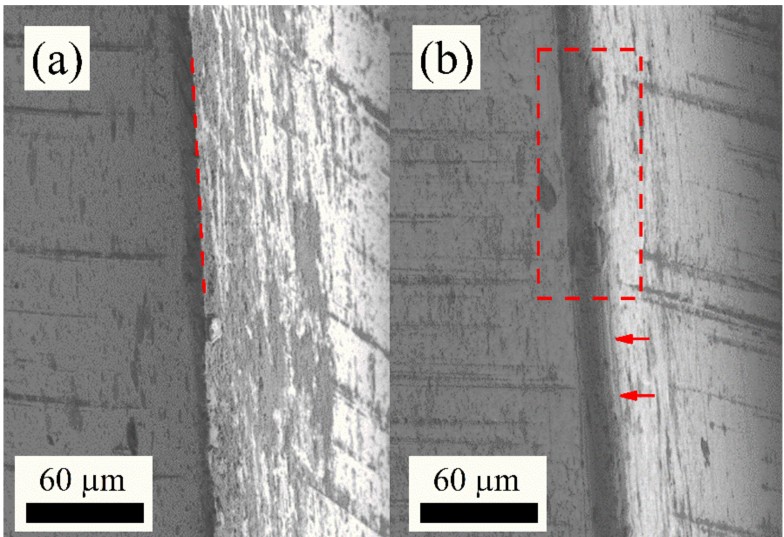

**Figure 5.** The sharp blades (**a**) made of 154CM stainless steel degraded into a round surface with abrasive grooves (**b**) after the cutting test.

The HRC hardness of these MSS samples were measured using digital Rockwell hardness tester (HRS-150), and at least five tests were conducted on one sample following standard procedure. The hardness of phases in the microstructure of these steels were determined utilizing a Brukers TI-980 Triboindneter (Minneapolis, MN, USA) equipped with a Berkovich indenter, and a maximum load of 3 mN was applied in these nanoindentation tests.

## 3. Results and Discussions

### 3.1. Phases and Microstructure

The chemical compositions of these steels were determined and summarized in Table 1. In all three MSSs, about 1 wt% carbon is incorporated in the alloys, resulting in carbon solid solution and significant amount of carbide precipitates. For the metallic elements, a high content of chromium (>12 wt%) is found in all three alloys: 14.93 wt% in 154CM, 16.52 wt% in 440C and 18.28 wt% in N690. To be noted, molybdenum is also found in the alloys: 3.68 wt% in 154CM, 0.93 wt% in N690, and 0.32 wt% in 440C. Both Cr and Mo are expected to ensure the alloys with strong corrosion resistance from acid and salt in environment [27].

**Table 1.** Composition of three martensitic stainless steels (wt%).

| Alloys | C | Si | V | Cr | Mn | Mo | Co | Ni | Fe |
|--------|------|------|------|-------|------|------|------|------|------|
| 154CM | 1.04 | 0.30 | 0.10 | 14.93 | 0.65 | 3.68 | 0.19 | 0.14 | Bal. |
| 440C | 1.02 | 0.35 | 0.13 | 16.52 | 0.49 | 0.32 | 0.17 | - | Bal. |
| N690 | 1.06 | 0.26 | 0.11 | 18.28 | 0.48 | 0.93 | 1.86 | 0.24 | Bal. |

Note: about 10% relative error exist in the composition of elements except for carbon due to semi-quantitative analysis on the bulk steel samples by XRF technique, and about 0.16% relative error exists in the carbon content.

According to XRD patterns in Figure 6, martensitic phase and $M_{23}C_6$ carbide phase could be identified in all of these MSSs. Though $M_7C_3$ is also expected to exist in the steels containing high Cr [17], the amount of $M_7C_3$ is probably too small to be detected, therefore its contribution to anti-wear property is negligible. The carbide precipitates embedded in steel matrix were imaged by backscattered electrons using SEM, as shown in Figure 7a–c. In contrast to bright steel matrix, carbide generally is darker due to the light element carbon in carbide that backscatters less electrons than heavier metallic elements. It is interesting to find the carbide phase turns from dark to bright in the sequence: 440C < N690 < 154CM. This phenomenon indicates the content of heavier metallic alloy elements in carbide phases

increase in the same sequence, which is confirmed by the elemental analysis of carbides in Table 2. Besides Cr (atomic weight 52) and Fe (atomic weight 55.8), much heavier Mo (atomic weight 95.94) was found most in the carbide phase in 154CM (11.53 wt% Mo), then less in N690 (3.28 wt% Mo) and none in 440C (0 wt% Mo).

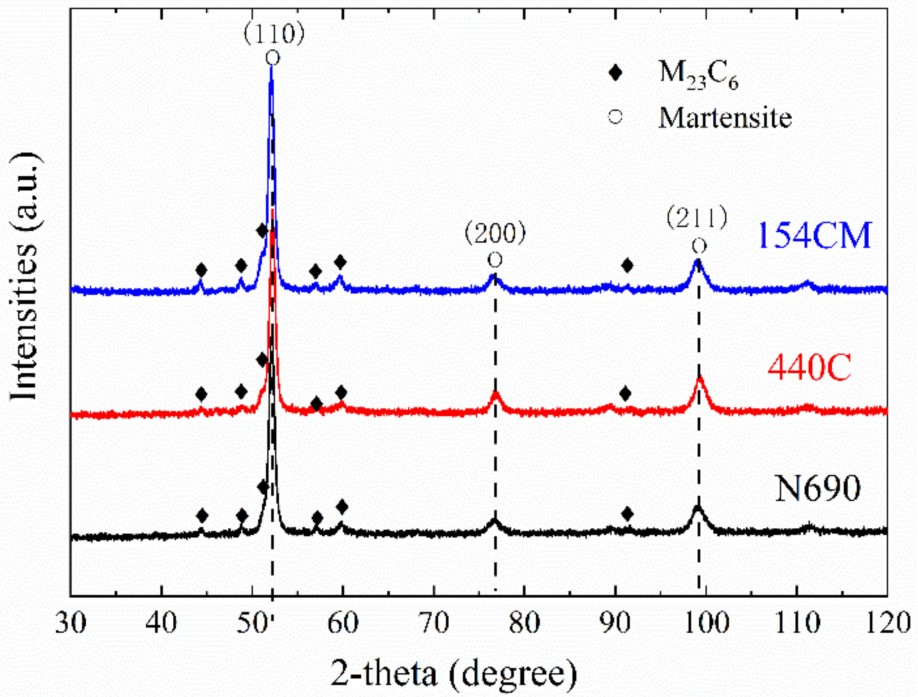

**Figure 6.** XRD patterns of the martensitic stainless steels for knife blade.

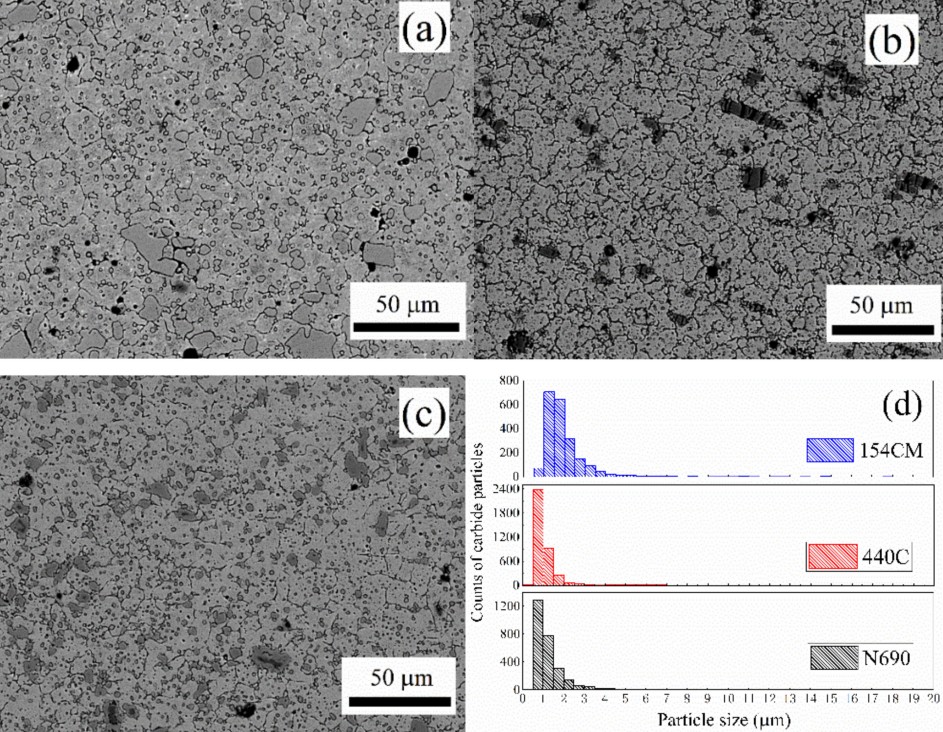

**Figure 7.** Microstructure of the martensitic stainless steels (**a**) 154CM, (**b**) 440C, and (**c**) N690 imaged by backscattered electron and the distribution of carbide particles in these three steels (**d**).

**Table 2.** Distribution of major metallic elements in steels (wt%).

| Phases in Steels | 154CM | | 440C | | N690 | |
|---|---|---|---|---|---|---|
| | Matrix | Carbide | Matrix | Carbide | Matrix | Carbide |
| Fe | 88.58 | 43.3 | 86.28 | 34.06 | 86.42 | 37.75 |
| Cr | 10.68 | 45.17 | 12.96 | 64.99 | 13.13 | 58.3 |
| Mo | - | 11.53 | - | - | - | 3.28 |
| Si | 0.74 | - | 0.77 | - | 0.45 | - |
| V | - | - | | 0.96 | - | 0.67 |

The quantity of carbide in steels were estimated from their area occupied in SEM image, and N690 contain about 13.03% carbide in area, more than those in 154CM (11.3%) and in 440C (7.86%). The size of carbide precipitates in these steels were measured and the size distribution were displayed in Figure 7d. Fine carbide precipitates scattered in the matrix of 440C have an average size of 1.07 ± 0.02 μm, while larger mean sizes of carbide particles are calculated as 1.38 ± 0.03 μm in N690 and 1.99 ± 0.03 μm in 154CM. The average spacing between carbide precipitates in steels are readily estimated from the counts of precipitates in the given area: about 3.4 μm for 440C, 3.9 μm for N690, and 4.5 μm for 154CM.

*3.2. Capability of the MSS Blades to Maintain Sharpness*

The cutting depth of paper cards per one cycle of blades made of 154CM, 440C and N690 were averaged and compared, as displayed in the Figure 8a. All of the data drop sharply in the initial several cutting cycles, and then decreases approximately at a stable rate. Since the paper cards break up underneath the blade as a result of the highly stressed zone created by the blade tip, and a higher stress results in larger cutting depth for one cycle. The continuous decreasing of cutting depth per cycle indicates the sharp blade tip continuously turns blunter in the cutting test, as displayed in Figure 8c. The stabilization of cutting rate is probably ascribed to a more stable geometry of blade tip achieved as the abrasive cutting proceeds. The persistence of blade cutting performance can be evaluated by the cumulative cutting depth of stacked cards in the test, as shown in Figure 8b, and the ICP and TCC of all of these three MSSs were calculated and compared in Figure 8d. Obviously, the blade made of 154CM exhibits significantly higher (about 25%) TCC than the other two blades made of 440C and N690. On the other hand, the ICP of these steels are quite closer to each other (<13.6%): 136.2 mm for 154CM, 129.4 mm for 440C and 119.8 mm for N690. Similar ICP performances confirm similar initial states of these blades, but steel type eventually plays a role as the cutting continues.

The appearance of blade edges made of three MSSs were inspected using SEM to investigate possible wear process of the blade edges in the cutting test. The characteristic cutting depth per cycle overlaps with the SEM image of the corresponding blade edge in Figure 9 to check possible correlation between abrasive feature on the blade tip and cutting performance. In Figure 9a, long grooves by silica are shown on a round smooth surface of the blade tip made of 154CM after cutting test, and the cutting depth per cycle also drops smoothly as the cutting test proceeds. In contrast, rather rough surface with peeling area (red circles) was observed for the blade tip of 440C after cutting test in Figure 9b, and small discontinuities (red arrows) were also found on the cutting depth per cycle for the same sample. In Figure 9c, some peeling pits (red circles) were observed on the worn blade tip of N690 after the cutting test, and larger jumps (red arrows) appear on the curve of cutting depth per cycle. There may exist a correlation between the wear process of the blade tips and instability of the CATRA cutting performance. When the carbide phase cracks and peels off from the blade tip, the blade tip loses hard and brittle particles that help cut up paper fibers, and therefore results in sudden discontinuities of cutting rate. The degree of instability in cutting seems to depend on the size of carbide pieces peeling away. For 440C with smoother blade edge, only small carbide pieces dropped, and small discontinuities occurred, while for N690 with rough blade tip, large jumps were observed,

since the cutting performance degraded more when larger carbides peeled off. The wear process got accelerated in the blades of 440C and N690, as blunter edge geometry was faster achieved due to peeling process, while for the blade of 154CM, scratch by abrasive particle seems to dominate the wear process of blade tip, and its sharpness was better maintained.

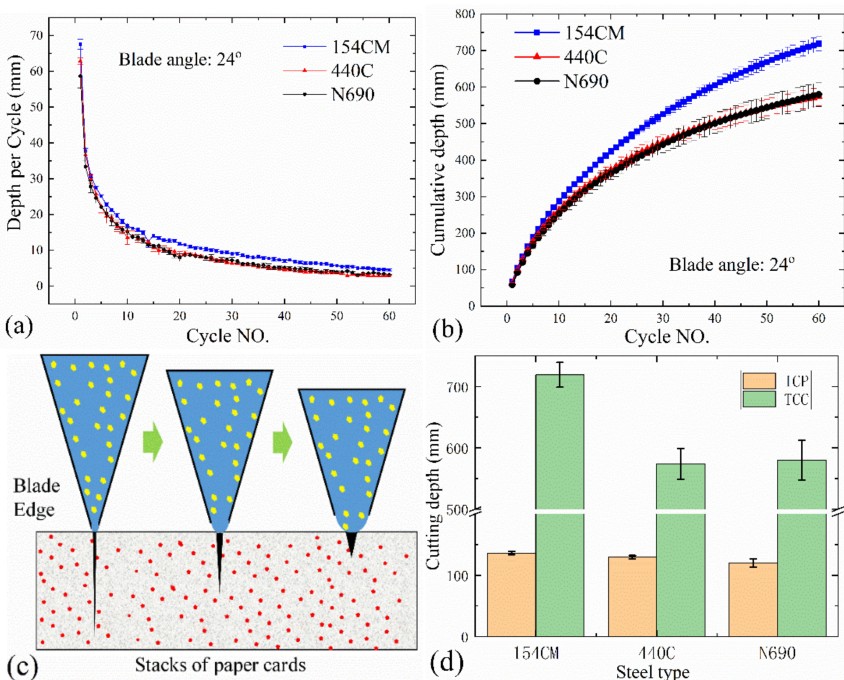

**Figure 8.** Cutting depth per each cycle (**a**) and cumulative cards cutting performance (**b**) of the MSS blades during the CATRA sharpness test revealed the degradation of blade sharpness (**c**), and ICP and TCC (**d**) of the MSS blades were calculated and compared.

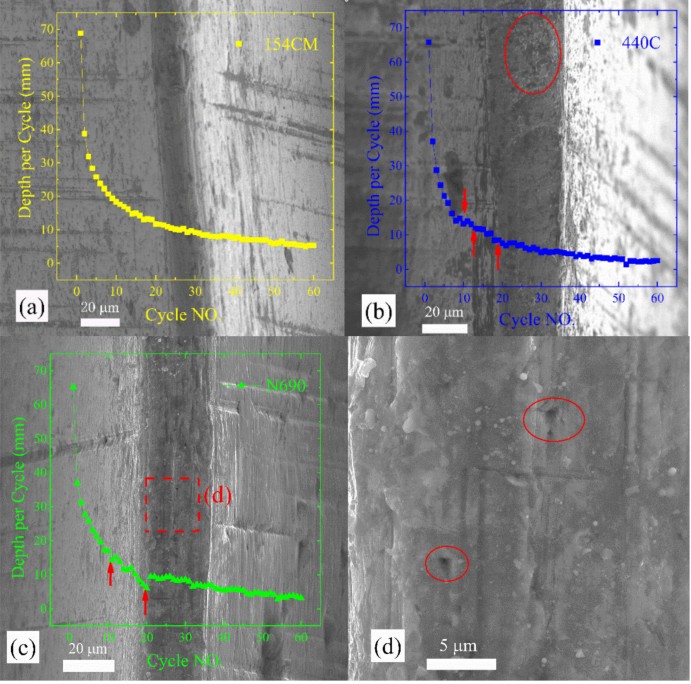

**Figure 9.** Surface of blade edge made of stainless steel 154CM (**a**), 440C (**b**), and N690 (**c**,**d**) after sharpness test overlapped with characteristic cutting depth per cycle for the corresponding steel blade.

*3.3. Hardness of Knife Steels: Rockwells (HRC) vs. Nanoindentation*

In the cutting tests, the abrasive silica particles reshape the blade tip into a blunter geometry, and the blunter blade geometry strongly deteriorates the cutting performance. Given the same initial blade edge geometry and the same sharpness test procedure followed, the discrepancy in the cutting performance of steel blades probably result from the mechanical properties of knife steels. Hardness is a crucial mechanical feature that quantifies the capability of materials to resist local plastic deformation, and it dominates the capability of steel blades to maintain their geometry under local stress during cutting. Generally, the plow mechanism describes the wear process of steel against hard brittle particles. According to the classic Archard's equation [28], the volume of material plowed away by asperities is

$$v = \frac{kW}{H}x \tag{1}$$

where $W$ is the normal load on the wear surface, $k$ is the non-dimensional coefficient that includes the geometry of the asperities (e.g., blade tip), $H$ is the hardness of materials, and $x$ is the distance of wear sliding. It indicates the volume of plowed-away or wear rate is inversely proportional to hardness of steels. The HRC hardness of 154CM, 440C and N690 were determined to be 57.2, 57, and 56.9, respectively. If the plow mechanism is the major wear scenario of steel blades, all of these three steel blades should exhibit the same cutting performance. However, it was found that 154CM is more superior to 440C and N690 in cutting stacks of paper cards, which is not surprising. As peeling found in 440C and N690 also contribute to the wear process of blade edges, and 154CM showed the best capacity to maintain the geometry of blade edge. To be noted, all of these martensitic steels contain numerous carbide particles with the range of size from less than 1 μm to over 10 μm. Since HRC hardness test gives an averaged hardness value from indents with several tens of micrometer, thus an area over hundreds of $μm^2$. Therefore, the HRC hardness may not be ideal to evaluate the blade wear processes that are mainly localized in sub-micrometer scale. In this study, the cutting performance of blades made of 440C and N690 are the same, even though the amount of carbide is significantly less in 440C than in N690. The finer carbides with dense distribution in 440C probably compensate the shortage of carbide quantity. Actually, steel matrix containing fine uniform carbides exhibits superior anti-wear performance [29,30] has been reported previously. Yet, more investigations are required on 154CM with fair amount of coarser carbide particles, as its supreme cutting performance is unexpected.

The hardness of martensitic matrix and $M_{23}C_6$ carbide phase in the MSS blades were determined using instrumented indentation technique. As displayed in the insets of Figure 10, the microstructures of steels were identified with sharp topological contrast in an area of 40 by 40 μm using scanning probe microscopy: the bright yellow isolated regions are carbide phases, and denser fine carbide particles were observed in 440C than in N690 and 154CM, while the dark brown regions are martensitic matrix, and green circles are marked for indentation sites. Notably, the brittle carbide phases are higher than the martensitic matrix in the steel sample, which indicates the carbides suffer higher pressure than matrix during mechanical polishing. Similarly, the carbides can also shadow the abrasive particles for the matrix in the sharpness test. For each phase of the steels at least 6 indentation tests were performed and the load-displacement (P-h) curves for these tests are shown in Figure 10. From the penetration depth, these P-h curves can be categorized into two groups: red curves for the hard carbide particles, and the black curves for the martensitic matrix.

According to the Oliver-Pharr method [31], the hardness and elastic modulus were calculated from the P-h curves of the indentation tests. The carbide phases in steels exhibit rather high hardness: 13.77~15.29 GPa, which matches the experimental [32] and calculation results [33] and agrees with the range of hardness: 9.5~14 Gpa for Cr-rich $M_{23}C_6$ carbide depending on its composition [34]. In contrast, the martensitic matrix of these

steels exhibits a lower range of hardness 6.6~7.37 Gpa, which are comparable with other martensitic steels [35,36], and much higher than austenitic steels [37]. It should be noted that larger variation in the hardness of steel matrix may result from its complicated structure that contains martensitic phases and residual austenite. As summarized in Figure 11a and Table 3, the martensitic matrix in all three MSSs: 154CM, 440C, and N690 were strengthened by harder carbide phases. The consistency in hardness could result in the discrepancy in cutting performance in steels.

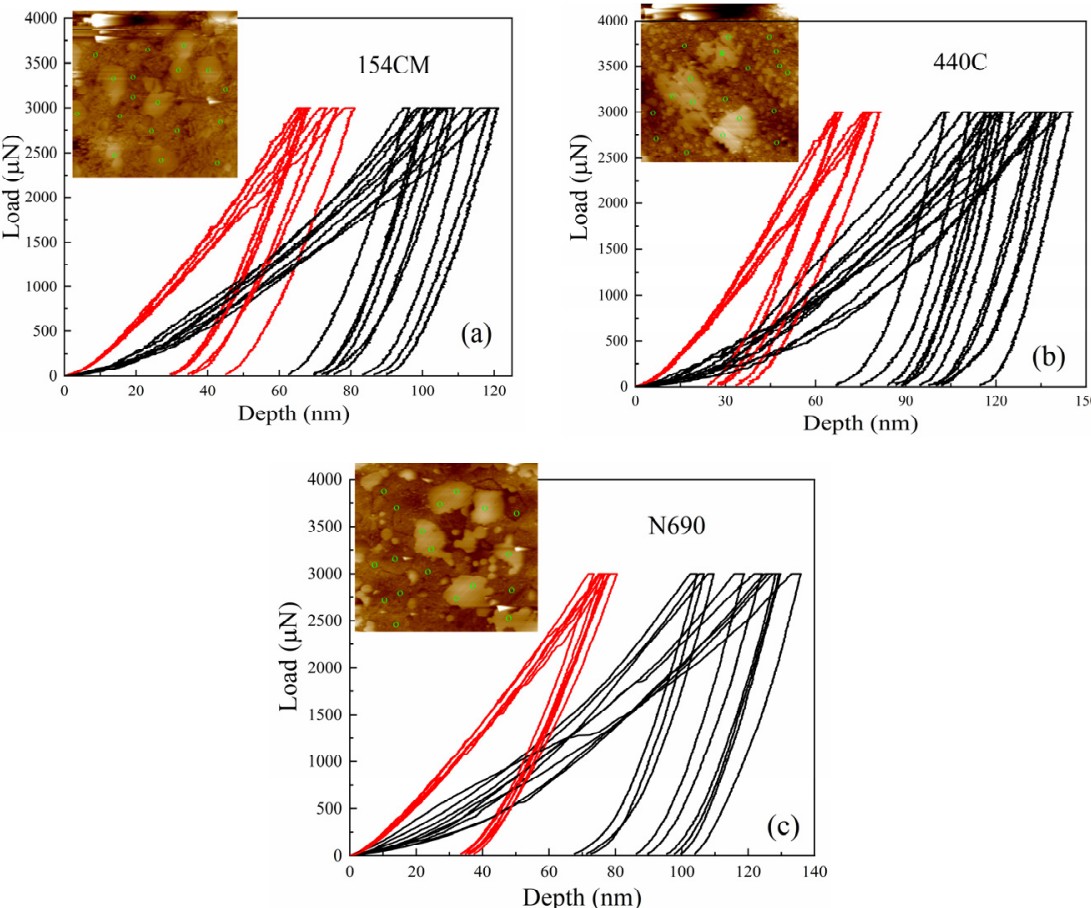

**Figure 10.** Load-displacement curves for nanoindentation tests on matrix and carbide M23C6 phases in stainless steels: (**a**) 154CM, (**b**) 440C, and (**c**) N690.

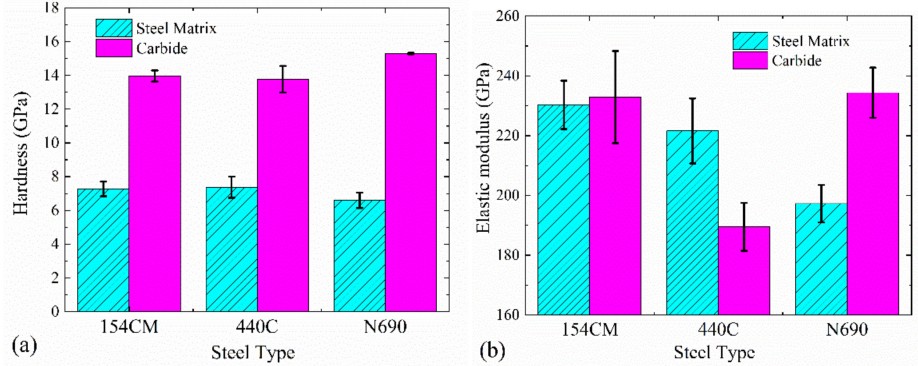

**Figure 11.** Comparison of hardness (**a**) and elastic modulus (**b**) of martensitic matrix and carbide phases in three stainless steels: 154CM, 440C and N690.

**Table 3.** Hardness and elastic modulus of martensitic stainless steel from indentation tests.

| Steels | HRC Hardness | Hardness (GPa) | | Elastic Modulus E (GPa) | |
|---|---|---|---|---|---|
| | | Matrix | Carbide | Matrix | Carbide |
| 154CM | 57.2 | $7.28 \pm 0.43$ | $13.97 \pm 0.33$ | $230.28 \pm 8.09$ | $232.90 \pm 15.43$ |
| 440C | 57.0 | $7.37 \pm 0.63$ | $13.77 \pm 0.79$ | $221.58 \pm 10.87$ | $189.47 \pm 8.00$ |
| N690 | 56.9 | $6.60 \pm 0.46$ | $15.29 \pm 0.04$ | $197.26 \pm 6.28$ | $234.35 \pm 8.42$ |

In terms of elastic response, the elastic moduli of phases are calculated from the relation [38]: $1/E_r = \left(1 - v_i^2\right)/E_i + \left(1 - v_s^2\right)/E_s$, where the reduced modulus $E_r$ is derived from the unloading response, $E_i = 1141$ GPa, $v_i = 0.07$ are elastic modulus and Poisson's ratio for diamond indenter [39], respectively, and $v_s = 0.3$ [40] is the Poisson's ratio for the steel sample. The elastic modulus of martensitic matrix for these steels were measured in the range of 197.26~230.28 GPa, which agree well with the modulus of martensitic steels [35], and the low value of modulus may result from the tempering state of martensite. The elastic moduli of carbide in these steels, on the other hand, show a significant discrepancy: 189.47 GPa for the $(Cr, Fe)_{23}C_6$ phase in 440C, while 234.35 GPa and 232.9 GPa for the $(Cr, Fe, Mo)_{23}C_6$ phases in 154CM and N690 steels, respectively. The lower modulus of $(Cr, Fe)_{23}C_6$ in 440C stands right between the calculated modulus 218.936 GPa for $Cr_{23}C_6$ and 186.365 GPa for $Fe_{23}C_6$ reported in literature [41]. Higher moduli of the carbide phases in 154CM and N690 may arise from Mo alloying. According to the calculation results [41], the introduction of Mo improves the strength and atomic interaction in the carbide system, and the calculated moduli of $Cr_{21}Mo_2C_6$ and $Fe_{21}Mo_2C_6$ can reach to 281.905 GPa and 279.849 GPa, respectively.

It should be noted that the elastic moduli of martensitic matrix and carbide phase in the steel 154CM were found to be rather close to each other, while significant mismatch in elastic modulus exists in both 440C and N690; the martensitic matrix shows higher elastic modulus than carbide in 440C, and the opposite trend was found in N690. It is proposed that the anomalous cutting performance of these steel blades is probably caused by the elastic modulus mismatch.

### 3.4. A Possible Wear Mechanism

In the sharpness test, a normal load of 50N is applied upon the test blade, and the stress between the blade tip and stacks of paper cards reaches to a significantly high level due to small contact area. The paper fibers underneath the blade tip break up at a high compressive stress beyond yield strength, and therefore the blade cuts through stacks of paper cards. In the cutting process as displayed in Figure 12a, micro-sized hard $SiO_2$ particles in the paper fibers rub the surface of steel blades, and plow away volume of steel materials. This abrasive wear occurs in all three steels, and gives rise to a blunt blade geometry.

Besides abrasive wear, the blade edge suffers fatigue damage due to the periodic load from the reciprocating movements as well as the vibration of blade edge during sharpness test. For the steels 440C and N690 with a modulus mismatch between matrix and carbide, the interaction force [42] proportional to shear modulus mismatch ΔG reaches a maximum at the carbide-matrix interface: $F_m = 0.05\Delta Gb^2(r/b)^{0.85}$, where $r$ is the radius of carbide particle, $b$ is the Burgers vector. The force weakens the interface bonding and aggravates the fatigue damage. Cracks nucleate and propagate in the steel matrix [43], and the carbide particles peel off easily from the martensitic matrix [17]. As a result, the steel matrix suffers more intensive abrasive wear without the shadow effect of carbides, and the deep abrasive grooves are formed as shown in Figure 12b. Eventually, the steel blade edge becomes more prone to get blunted. In the case of 154CM with similar moduli of matrix and carbide, the carbide-matrix interface stays robust, and the blade edge is more capable of maintaining a sharp geometry (even though the fatigue conditions remain).

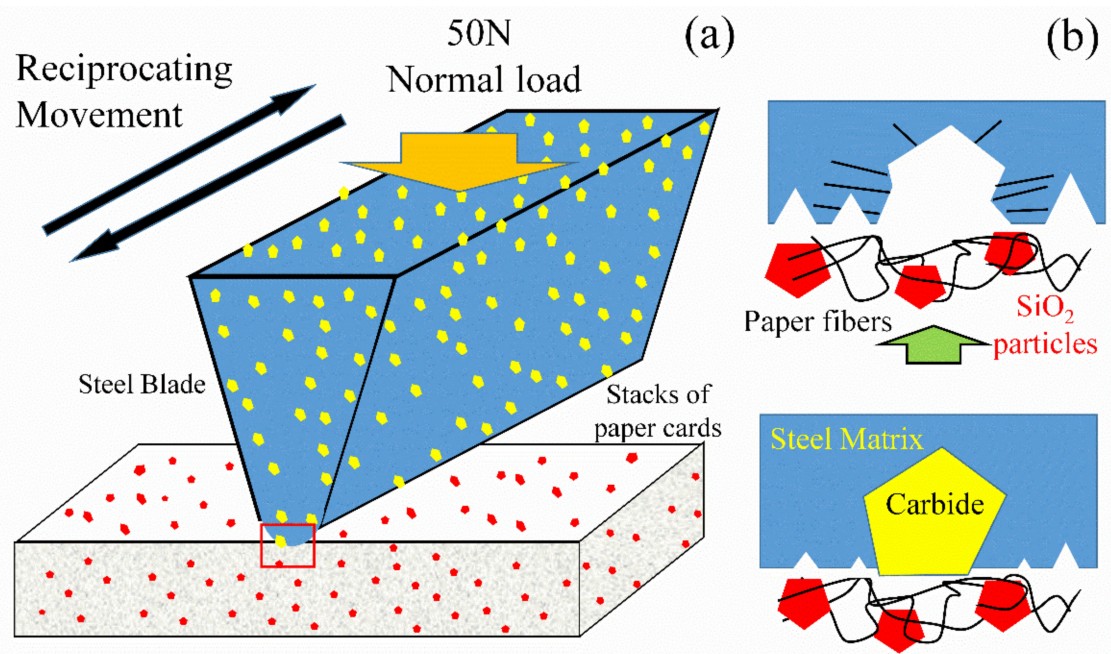

**Figure 12.** Schematic diagram (**a**) for a possible wear mechanism of steel blade edge, and the enlarged view of blade tip (**b**) displays small scratches on the steel matrix in the presence of carbides, while the cracks and deep abrasive grooves appear in steel matrix after the carbides peel off due to cooperation of abrasive wear and fatigue wear in the sharpness test.

## 4. Conclusions

In the study, steel blades were fabricated from heat treated martensitic stainless steels 154CM, 440C, and N690. The sharpness and its retention capability of all steel blades were investigated and compared based on an ISO international standard. Main findings are summarized below:

1.  The blade made of 154CM exhibited better sharpness retention than 440C and N690, while the HRC hardness values of these steels were almost the same. Conventional plow mechanism for abrasive wear is unable to explain the discrepancy in cutting performance.
2.  It was found that the hardness of martensitic matrix in MSSs fall in the range of 6.6~7.37 GPa, while the hardness of carbide phases reaches to a higher range: 13.77~15.29 GPa. The discrepancy in sharpness retention is unlikely to result from the similar hardness results of the MSSs.
3.  Mo content seems to strengthen the atomic interactions in carbides and raise the elastic modulus significantly. According indentation results, the elastic moduli of carbide (232.9 GPa) and martensitic matrix (230.28 GPa) match each other in 154CM, while elastic modulus mismatch between matrix and carbide was observed in 440C and N690, respectively.
4.  The modulus mismatch between the matrix and carbide results in additional force at the matrix-carbide interface, and therefore weaken the stability of carbide in steels. Fatigue peelings were more likely to occur in 440C and N690, and gave rise to easily-worn blade edge. The blade made of 154CM, however, mainly suffered abrasive wear by $SiO_2$ particles and exhibited the best capability of sharpness retention.

**Author Contributions:** Conceptualization, Q.Z.; methodology, W.L.; formal analysis, D.W. and Q.Z.; investigation, D.W.; resources, W.L.; data curation, D.W.; writing—original draft preparation, D.W.; writing—review and editing, D.W. and Q.Z.; supervision, Q.Z.; funding acquisition, Q.Z. All authors have read and agreed to the published version of the manuscript.

**Funding:** This work was supported by "the Fundamental Research Funds for the Central Universities (WUT:2021IVA001)" and sponsored by Yangjiang Science and Technology Bureau under "Key Science and Technology Special Project (SDZX2020002)".

**Institutional Review Board Statement:** Not applicable.

**Informed Consent Statement:** Not applicable.

**Data Availability Statement:** The data presented in this study are available on request from the corresponding author. The data are not publicly available due to privacy.

**Conflicts of Interest:** The authors declare no conflict of interest.

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
