# Peer review of "Effect of Alloying Elements on the Sharpness Retention of Knife Blades Made of High Carbon Martensitic Stainless Steels"

_metals, doi:10.3390/met12030472_

Round 1
Reviewer 1 Report
The reviewed paper titled “Impact Effect of alloying elements on the sharpness retention of knife
blades made of high carbon martensitic stainless steels” is of a high scientific level. The relationship between the chemical composition and the content of alloying elements as well as the fatigue strength and sharpness of knives was demonstrated. The authors did not avoid minor errors and inaccuracies. Notes for reflection and minor correction of the work:
- Figure 1 - no description of the heating and cooling rate in the drawing
- - there is not enough information at work on the preparation of samples for metallographic tests, whether the samples were etched, if so, with what reagent
Author Response
Responses:
The authors feel grateful to the reviewer for the precious suggestions. In the revised manuscript, we have added description as suggested. Thanks! Below are our point to point responses:
- In Figure1, we have added the heating and cooling rates.
- The technical details about sample processing for microstructure observation have also been added in the text body of manuscript.
Sorry for missing the technical information. We are looking forward to your positive feedback. Thanks!
Prof. Qinyi Zhang

Reviewer 2 Report
The article concerns the improvement of the durability of knife blades through material changes based on wear tests. The article is interesting but requires some additions. The most important considerations are:
1. Add schema or view a testing machine developed by CATRA
2. Table 1 should provide the accuracy of the determination of the chemical composition and the device used for its determination
3. Rows 176-179 In addition to the mean value, the measurement uncertainty should be given
4. Line 180 Enter text between the subsection title and Fig. 7
5. Fig. 7d Steel markings not very clear, the type should be enlarged
6. Fig. 8 not very clear, the surface view should be improved
7. Line 311 Enter text between the subchapter title and Fig. 11
8. The wear model in Fig. 11 is not accurate, the stages of wear after carbide removal should also be shown, it is not known whether the wear is due to fatigue, brittle or scratching processes.
9. At the beginning of the article, the authors should clearly indicate the novelty of the work.
Author Response
Responses:
The authors are grateful to the reviewer for the precious suggestions, and feel sorry to miss some important information in the manuscript. Here the manuscript has been revised accordingly. The responses to the reviewer’s suggestions are listed below:
- We have added a photo with schematic diagram (Figure 3) to illustrate the cutting test.
- The devices used to determine the chemical composition have been included in the manuscript. We didn’t obtain specific uncertainty for each element, but generally the accuracy of elemental contents semi-quantitatively determined by XRF is fair (relative error <10 %), so we added a footnote in Table 1. Though the error is a bit high, we can still compare and judge the content of Mo and Cr in steels qualitatively.
- Uncertainty has been added to the size of carbides in steels.
- We have arranged the text as suggested.
- We enlarged the fonts of the Steel Markings in Fig.7d and also in Fig. 6d, Fig 10a and 10b
- We improved the DPI of those figures, especially brightened (c) and (d) for more clear details of scratches and pits due to carbide peeling.
- We have arranged the text as suggested.
- Thank you for your comments. Yes, the wear model we proposed is not an accurate one. The wear process of steel blade is complicated, most likely both fatigue and scratch contribute to the wear. Nonetheless, we still provided a viewpoint to explain the observed facts including cutting performance, images of worn blade as well as composition and hardness of the phases in blade steels. In our opinion, the wear process of blade steel is continuous: some carbide peels off from the blade body, at the same time fresh carbide may expose to the paper cards, so we don’t think there is stage of wear “after carbide removal”, therefore, we would like to keep current schematic diagram.
- We have added relevant statement to describe the significance of this work in the introduction part. Thanks!
We are looking forward to your positive feedback. Thanks!
Prof. Qinyi Zhang

Reviewer 3 Report
REVIEW COMMENTS
Manuscript ID: metals-1615194
Effect of alloying elements on the sharpness retention of knife blades made of high carbon martensitic stainless steels
The authors analyze the differences in sharpness retention of knife blades made of three martensitic stainless steels. For this, they manufacture knife blades with the same geometry and carry out standardized wear tests. In these tests, they observe that one of the steels has a better behavior than the other two steels. The authors observe that the wear mechanism is different in the steels that show worse behavior. In the three steels, it is observed that abrasive wear occurs on the edge of the knife blades in the wear tests. However, in the two steels that have the worst sharpness retention, it is observed that spalling also occurs on the surface of the knife blades. The authors carry out a detailed analysis in order to explain the differences in the wear mechanism. They measure the HRC hardness, the microhardness of the matrix and carbides and calculate the elastic modulus of the matrix and carbides. The results of the hardness and microhardness tests do not explain the differences in the wear mechanism. However, if they observe differences in the elastic modulus of the matrix and the carbides of the steels in which a spalling wear mechanism is observed. The authors propose that these differences in the elastic modulus between the matrix and carbides can induce a surface fatigue mechanism that facilitates the formation of cracks that produce the observed spalling.
The work is interesting and, for the most part, clearly explained. In any case, I would appreciate it if you would consider the following suggestions:
- Lines 53-54: "annealing at higher temperature and then fast quenching". This sentence is imprecise. Annealing is a heat treatment that can have the objective of softening a steel to, for example, increase its ductility. Never harden it. To harden a steel by martensitic transformation, the steel must be austenitized. Correct it, please.
- Line 54: It must be indicated in advance what the acronym MSS means. It is indicated later in the line 60. Correct it, please.
- Line 62-63: Is it shown in references 21 and/or 22 that the addition of Ni improves pitting resistance? In many works it is shown that Ni does not improve resistance to pitting and is not a chemical element that is part of the PREN (Pitting Resistance Equivalent Number). Check it, please.
- Lines 83-86: What is the thickness of the heat treated steel plates? Indicate it , please.
- Line 101: “One longitude edge”. What are the authors referring to? I do not understand this phrase. Correct it, please.
- Line 117: Typographic error. "Stroke" instead of "Stoke". Correct it, please.
- Line 128: The images are difficult to interpret. They should be explained in more detail in the text, indicating the different areas in the same image or the most important characteristics.
- Line 198: The authors indicate that the 154CM steel has a higher ICP value than the other two steels. However, in Fig. 7(d) it appears that the values are similar for the three steels. It may be clearer if the values for the three steels and their percentage difference are indicated.
- Line 202: It would be interesting to include higher quality images where abrasive wear marks and spalling zones can be more clearly observed.
- Line 209: In Fig. 8 the microstructure of the steels is not shown. Therefore, the possible correlation between microstructure feature and cutting performance is not verified. The microstructure is shown in Fig. 6. Please correct it.
- Line 214 and Line 216: What is the cause of the small discontinuities and large jumps in the curves of depth per cycle? Explain it clearly, please.
Author Response
Responses:
The authors appreciated to the precious informative suggestions raised by the reviewer, as we believe they can help improve the quality of our work. Please review the manuscript revised accordingly. Below listed our responses to the reviewer’s suggestions:
- Pointed out the purpose of annealing as suggested.
- Define the acronym MSS when it appears for the first time.
- Thank you for point out that. Yes, Ni is not a typical element that improves the resistance to pitting. we decided to remove Ni in our statement to avoid confusion.
- Add the thickness of steel plate in the manuscript.
- Rewrite that sentence. Sorry for confused description.
- Corrected it, sorry for the typo.
- Mark the SEM image as suggested, and described the major difference between before and after cutting.
- Rewrite this part for more clear discussion
- We improved the DPI of those figures, especially brightened (c) and (d) for more clear details of scratches and pits probably due to carbide peeling.
- Clarified the description. Sorry for using incorrect concept.
- The cause of small discontinuities and large jumps were explained in the main body.
We are looking forward to your positive feedback. Thanks!
Prof. Qinyi Zhang

Round 2
Reviewer 2 Report
The model of wear of the blade is an important part of the article. It must be as appropriate as possible to the tests carried out. The authors have to rethink it once more and correct it so that it shows the kinetics of damage development, and not just a certain intermediate stage. I have no objections to the rest of my comments, they have been corrected.
Author Response
Response:
The authors agree with the reviewer’s concern about the wear model of the blade edge and appreciate the reviewer’s comments. In the revised manuscript, we modified the enlarged view of blade edge with/without the carbide. Generally, the blade tip suffers fatigue wear and abrasive wear at the same time. For carbide, the periodic load and collisions with SiO2 particle result in the peeling-off issue, on the other hand, the steel matrix shadowed by carbides suffer less impact from the paper fibers with SiO2 particles, therefore only small abrasive grooves appear near the carbides. When the carbides peel off, it brings the steel matrix alone under the attack of SiO2 particles, and the scratches grow into deep abrasive grooves on the steel matrix without the shadow effect of carbides.
We introduced the shadow effect of carbides in the scanning image of steel microstructure (Fig. 10). In the schematic diagram, we also modified the size of SiO2 particles, as they should be closer to size of carbides according to the SEM images.
We are looking forward to your positive feedback. Thanks!
Prof. Qinyi Zhang

Reviewer 3 Report
REVIEW COMMENTS
Manuscript ID: metals-1615194-v2
Effect of alloying elements on the sharpness retention of knife blades made of high carbon martensitic stainless steels
The manuscript has significantly improved. I am grateful that the authors have taken into account the suggestions made by this reviewer. Some images have been improved and the evidence of the wear mechanism is now more clearly observed. The description in the main body of the manuscript has also improved and it is now possible to understand the work more easily than in the initial version. However, in my opinion, there are still some details that should be corrected. I would appreciate if you would consider the following suggestions:
:
- Line 54: The authors refer to the treatment as "annealing" and this is incorrect. Annealing is a heat treatment that can have the objective of softening a steel to, for example, increase its ductility. Never harden it. To harden steel, samples must be soaked at high temperature to fully austenitize them, followed by rapid cooling. This treatment is called quenching, not annealing. Correct it, please.
- Line 144: In Figure 5a the green dotted line is not clearly visible. Please use a thicker line or a line of another color.
- Line 144: In Figure 5b indicate with arrows the shallow grooves scratched by the SiO2 particles. This makes it easier for the reader to interpret the images.
- Line 259: In Fig. 8 the microstructure of the steels is not shown. Figure 8 shows the appearance of blade edges. Please correct the text.
Author Response
The authors appreciated the reviewer’s comments that help improve the quality of our manuscript. We have revised the manuscript as suggested, please see our response point to point below:
- We mean “anneal” to be a regular heat treatment that keep the steel sample at a certain higher temperature for a planned period of time. This concept may confuse reader when we discuss it together with quenching. We corrected the statement into a straightforward form in the introduction part.
- Thanks for pointing out that. We have modified the Figure 5a and 5b as suggested. For a clear display, thicker red dotted line and red arrows replaced the green ones. At the same time, we also modified the description of Fig.5 in the text.
- We have reviewed the discussion about Fig.9 (former Fig 8 with SEM images of blade ends) once again and corrected any misuse of “microstructure”. Sorry for our overlooking mistakes.
Thank you again for your nice comments on our manuscript. Look forward to your positive feedback.
Prof. Qinyi Zhang
